# Immune Regulatory Processes of the Tumor Microenvironment under Malignant Conditions

**DOI:** 10.3390/ijms222413311

**Published:** 2021-12-10

**Authors:** Katrin Pansy, Barbara Uhl, Jelena Krstic, Marta Szmyra, Karoline Fechter, Ana Santiso, Lea Thüminger, Hildegard Greinix, Julia Kargl, Katharina Prochazka, Julia Feichtinger, Alexander JA. Deutsch

**Affiliations:** 1Division of Hematology, Medical University of Graz, Auenbruggerplatz 38, 8036 Graz, Austria; katrin.pansy@medunigraz.at (K.P.); barbara.uhl@medunigraz.at (B.U.); marta.szmyra@medunigraz.at (M.S.); ka.fechter@medunigraz.at (K.F.); lea.thueminger@stud.medunigraz.at (L.T.); hildegard.greinix@medunigraz.at (H.G.); KatharinaTheresa.Prochazka@uniklinikum.kages.at (K.P.); 2Division of Cell Biology, Histology and Embryology, Gottfried Schatz Research Center for Cell Signaling, Metabolism and Aging, Medical University of Graz, Neue Stiftingtalstraße 6/II, 8010 Graz, Austria; jelena.krstic@medunigraz.at (J.K.); julia.feichtinger@medunigraz.at (J.F.); 3Division of Pharmacology, Otto Loewi Research Center, Medical University of Graz, Universitätsplatz 4, 8010 Graz, Austria; ana.santiso-sanchez@medunigraz.at (A.S.); julia.kargl@medunigraz.at (J.K.)

**Keywords:** tumor microenvironment, anti-tumor immune responses, immune evasion, immune checkpoint, cytokines, metabolism, antigen presentation, TME targeting therapy

## Abstract

The tumor microenvironment (TME) is a critical regulator of tumor growth, progression, and metastasis. Since immune cells represent a large fraction of the TME, they play a key role in mediating pro- and anti-tumor immune responses. Immune escape, which suppresses anti-tumor immunity, enables tumor cells to maintain their proliferation and growth. Numerous mechanisms, which have been intensively studied in recent years, are involved in this process and based on these findings, novel immunotherapies have been successfully developed. Here, we review the composition of the TME and the mechanisms by which immune evasive processes are regulated. In detail, we describe membrane-bound and soluble factors, their regulation, and their impact on immune cell activation in the TME. Furthermore, we give an overview of the tumor/antigen presentation and how it is influenced under malignant conditions. Finally, we summarize novel TME-targeting agents, which are already in clinical trials for different tumor entities.

## 1. Introduction

The tumor microenvironment (TME) consists of immune cells, non-immune stromal cells, and extracellular matrix proteins. All of these components function as critical regulators of tumor growth, progression, and metastasis as well as anti-tumor immune responses. Recently, treatment options for various tumor types have changed tremendously with the development of immunotherapy. This novel therapy differs from conventional chemotherapeutic agents in that it reactivates the immune responses to malignant cells, rather than directly affecting tumor cell growth, pointing out the importance of the TME, especially of immune cells, in tumors. Immune evasion is a mechanism that prevents the immune cell-mediated tumor cell lysis and it has become an important research field in various tumor entities in the last few years. This complex mechanism is mediated by multiple interacting molecular processes, including: (1) suppression of the lymphocyte activation and effector functions mediated by direct interaction of tumor and immune cells and/or the active recruitment of cells possessing immunosuppressive properties and (2) antigen presentation (Figure 1).

Here, we review how the TME and its interaction with tumor cells and/or the immune system affect immune evasion, which regulators and pathways are involved, especially focusing on immune cell activation and antigen presentation.

## 2. Composition and Functions of the TME

The TME consists of more than 30 distinct tumor-infiltrating non-malignant types of cells, as well as the extracellular matrix (ECM), and its composition differs among tumor types [1,2,3,4,5,6]. The cellular component comprises immune cells of the innate immune system (neutrophils, eosinophils, macrophages, dendritic cells (DCs), mast cells, and natural killer (NK) cells, as described in detail in Table 1) and the adaptive immune system (T and B cells, as described in detail in Table 2 and Table 3), fibroblasts, endothelial cells, and various tissue-associated cells. These cells interact with each other and with cancer cells via complex communication networks through secreted cytokines, chemokines, growth factors, and proteins of the ECM (fibrous structural network, glycoproteins, growth factors, and proteoglycans). The TME cells and the ECM can have a tumor-promoting function but can also play a key role in the regulation of anti-tumor immune responses.

The immune cells of the TME can possess both tumor-promoting as well as anti-tumor functions in the TME (summarized in Table 1, Table 2 and Table 3). The key player in immune cell-mediated tumor rejection is the subgroup of cytotoxic T cells (CD8+ T cells, Table 3).

A subpopulation of cancer-associated fibroblasts (CAFs) possesses the ability to modulate anti-tumor immune responses. This function is mainly mediated by the secretion of anti-inflammatory cytokines, prostaglandins (e.g., PGE-2), chemokines, and ECM proteins [4,5,6]. CAFs can increase the number of Tregs by the secretion of chemokines and cytokines. Furthermore, CAFs can directly inhibit the activity of effector T cells and cytotoxic T cells (CD8+ T cells) by the secretion of anti-inflammatory cytokines and PGE-2, by the expression of immune inhibitory ligands, such as PD-L1 and PD-L2 (described in detail in Section 3.1.3 and Section 3.1.4) and/or by the production of metabolic reprogramming factors. The immune inhibitory effects of CAFs can also be promoted by their production of dense collagen networks, which represent a physical barrier that inhibits the presence of activated T cells within the tumor. ECM remodeling mediated by CAFs can also acts on the macrophage differentiation into the M2 anti-inflammatory phenotype, thereby playing a key role in their immune inhibitory functions.

Endothelial cells in the TME play a significant role in the pathogenesis of different types of tumors. They are crucial for angiogenesis and are therefore needed for delivering nutrients to tumor cells, promoting tumor cell survival, and aiding in metastasis [8]. Moreover, a study indicated that pericytes of the TME vessels can induce anergy of the CD4+ T cells through a regulator of G-protein signaling-5 (RGS-5)- and IL-6-dependent pathways [9].

## 3. T Cell Function in Malignant Conditions

### 3.1. T Cell Activation

T cells play a key role in immune cell-mediated tumor rejection. An antigen-presenting cell (APC) is needed for T cell activation. The APC bears antigenic peptides, which are non-covalently bound to a major histocompatibility complex (MHC) class I or class II molecule. Under malignant conditions, the antigenic peptides are tumor-specific antigens, e.g., neoantigens arising from somatic mutations and/or cancer-associated antigens. The TCR only recognizes the tumor-specific antigen when it is bound to the MHC molecule. In T cells, the TCR associates with the CD3 membrane complex, whose cytosolic region is responsible for propagating the intracellular signal upon TCR ligation. Each TCR is associated with either a CD4 or CD8 co-receptor, which both bind to MHC (class I for CD8 and class II for CD4) to further stabilize the T cell and APC interaction [10]. Besides the TCR-MHC interaction, other relevant molecules (co-stimulatory and/or adhesion) are needed for T cell activation. The primary co-stimulatory pathway is the binding of CD28 expressed on T cells and CD80 and/or CD86 expressed on APCs [11]. However, additional membrane-bound and/or membrane-soluble signals are necessary for a complete T cell activation. These signals are achieved by the TNF receptor superfamily (CD27, OX-40, 4-1BB, and CD30) when interacting with their appropriate ligand on APCs (CD70, OX-40L, 4-1BBL, and CD30L). The culmination of the TCR and co-stimulatory signals cause the production/secretion of IL-2, autocrine and paracrine factors that stimulate T cell proliferation [12,13,14]. T cells also express a set of inhibitory receptors, called immune checkpoints, which are implicated in the fine-tuning of inflammatory processes. These inhibitory receptors can limit the co-stimulatory signal as well as the co-stimulatory molecule ligation and are needed for T cell homeostasis [15]. Under malignant conditions, T cell activation and effector function are inhibited by soluble factors, e.g., cytokines, PGE and toll-like receptor (TLR) ligands, by inhibitory receptors/immune checkpoints, and/or by cells possessing regulatory functions that result in T cell dysfunction, e.g., Tregs, M2 macrophages and myeloid-derived suppressor cells.

#### 3.1.1. T Cell Dysfunction in the TME

##### T Cell Anergy

T cell anergy is described as the induced hyporesponsive state of T cells caused by incomplete activation and/or low IL-2 production, resulting in little to no T cell proliferation. Human tumor cells and tumor-associated APCs often express high levels of PD-L1, PD-L2, ICOS-L, and B7-H3, which causes high co-inhibitory signals that are accompanied by a low CD28 co-stimulation of T cells, which is itself caused by low to absent CD80 and/or CD86 signals [16,17,18,19]. Consequently, the factors that negatively regulate TCR-coupled pathways, together with the epigenetic mechanisms that cause transcriptional silencing, are activated in T cells under these circumstances [20]. The T cell anergy process is mediated by the combination of the nuclear factor of activated T cells (NFAT) homodimer formation, members of the E3 ubiquitin family, which are also known as casitas B-lineage lymphoma (CBL), epigenetic factors such as Ikaros family zinc finger protein 1 (IKAROS), and the regulatory protein SIR2 homologue (Sirt1) [21,22,23]. 

##### T Cell Exhaustion

Exhausted T cells are described as effector T cells with decreased cytokine expression and effector cell function, as well as being re-activation resistant. This dysfunction occurs when T cells are chronically stimulated and/or activated in chronic inflammation and cancer [24]. T cell exhaustion seems to be a progressive process in which T cells experience repeated activation. As a result, T cells acquire multiple inhibitory surface molecules in persisting disease conditions such as cancer [25,26]. In cancer, the expression of the programmed cell death protein 1 (PD-1), T cell immunoglobulin domain and mucin domain-containing protein-3 (TIM-3), lymphocyte activation gene 3 (LAG-3), T cell immunoglobulin and ITIM domains (TIGIT), and B and T lymphocyte attenuator (BTLA) are found on exhausted T cells [25,27,28]. This results in reduced IL-2, IFN-γ, and TNF-α secretion as well as cell cycle arrest, which defines immune cell dysfunction [26]. Notably, it is also possible that exhausted T cells may express multiple inhibitory surface molecules [28,29,30,31]. They have a unique expression signature with alterations in the TCR- and cytokine-signaling pathways, as well as the signaling pathways that are implicated in migration and metabolism, and in the expression of transcription factors (B lymphocyte-induced maturation protein-1 (Blimp-1), basic leucine zipper transcription factor (BATF), NFAT, T-box expressed in T cells (T-bet), and T-box brain protein 2 (Eomes)) [24,32,33,34,35,36]. 

##### T Cell Senescence

Senescent T cells are characterized by telomere shortening, loss of co-stimulatory molecule CD28, and cell-cycle arrest [37,38]. Telomere shortening is a by-product of cell division and causes senescence [39]. Senescent T cells develop defective T cell effector abilities with decreased expression of IFN-γ, granzyme B, and perforin [40,41,42,43] and negative regulatory function [44,45]. Compared to T cell exhaustion and anergy, T cell senescence is irreversible [40,46,47]. T cell senescence can be induced by tumor cells via direct interaction, causing the activation of p53, p21, and p16, or by glucose competition and/or hypoxic TME [48].

In addition to low CD28 expression, T cell senescence is associated with low expression of CD27 and high expression of TIM-2, CD57, and killer cell lectin-like receptor G1 (KLRG-1), as well as p26 and p21, which are implicated in cell cycle regulation [26,49,50,51,52,53]. Moreover, the levels of other immune inhibitory surface molecules, e.g., TIGIT and CTLA-4, are increased on senescent T cells [48]. T cells with this surface molecule-expression pattern were found in melanoma [52] and lymphoma [48]. A recent study reported that IL-7 and its receptor also play a key role in immunosenescence [48].

#### 3.1.2. Immune Checkpoints/Inhibitory Co-Stimulatory Signals Causing Immunosuppressive Conditions in the TME

Immune checkpoints and their ligands (the most important members are depicted in Figure 2) can be subdivided into the B7-CD28 families, TIM family, immunoglobulin (Ig) superfamily, nectin and nectin-like binding receptors, and butyrophilins, as well as intracellular immune checkpoints. Blocking various regions of these immune checkpoint axes by therapeutic antibodies, which is referred to as immune checkpoint blockade (ICB), represents a novel and promising strategy that has proven its worth in clinical practice for different types of cancer, at least for CTLA-4 and PD-1 blockades [54].

Within this review, we focus on different types of immune checkpoints/inhibitory co-stimulatory molecules, their regulation, and their key function in the suppression of anti-tumor immune responses.

##### Co-Inhibitory B7-CD28 Family Members

The B7-CD28 family comprises six CD28 receptor family members and ten ligand members consisting of co-stimulatory or co-inhibitory molecules, which play an essential role in T cell tolerance and homeostasis (Figure 2) [55]. The B7 family co-inhibitory pathway includes CD80/CD86-CTLA-4, PD-L1/PD-L2-PD-1, HVEM-BTLA, as well as VISTA-axes, all of which are implicated in the mediation of the immune evasion of malignant cells [56].

One of the most studied inhibitory receptors is CTLA-4, which has structural and biochemical similarities to the co-stimulatory receptor CD28. CTLA-4 is the inhibitory counterpart of CD28, which shares the same ligands CD80 and CD86 [57]. CTLA-4 competes with CD28 for CD80/CD86 binding with a much higher affinity than CD28 in order to reduce T cell responses via cell-intrinsic and -extrinsic pathways [57,58]. In tumorigenesis, CTLA-4 binding to CD80/CD86 dampens T cell activation by blocking downstream PI3K/AKT and NF-κB pathways [56,59,60].

Another well-characterized member of the B7-CD28 family is PD-1. Its expression is caused by persistent exposure to antigens and/or to inflammatory signals during chronic infection [33,61,62]. PD-L1, the corresponding ligand, is broadly expressed on numerous immune cells such as (activated) T cells, B cells, DCs, and many cancer cells under malignant conditions to suppress T cell responses. The PD-L2 expression as the second PD-1 ligand is restricted to APCs [63,64,65]. The expression of PD-L1 and, to a lesser degree, of PD-L2, is induced in response to pro-inflammatory cytokines such as TNF-α and type I and type II IFNs [66,67,68]. Moreover, PD-L2 expression is upregulated through IL-4, granulocyte-macrophage colony-stimulating factor (GM-CSF) as well as interferons [69,70]. The binding of PD-L1 or PD-L2 to PD-1 weakens T cell activation and lowers the activation of immune cells by reducing the activation of intracellular signaling pathways and downregulation of effector cytokines [63]. In malignant conditions, it has been observed that PD-1-ligands prevent tumor cells from immune cell-mediated lysis [71].

The inhibitory receptor B and T cell lymphocyte attenuator (BTLA) is another member of the B7-CD28 family [72]. The primary known ligand is the Herpes virus entry mediator (HVEM), which is a TNF receptor (TNFR) family member. HVEM may also act as a ligand for CD160, tumor necrosis factor superfamily member 14 (LIGHT/TNFSF14), another TNFSF ligand, and Lymphotoxin α (LTα). The interaction of BTLA and/or CD160 with HVEM leads to inhibitory signals and causes the suppression of T cells. In contrast, BTLA binding by LIGHT and/or LTα results in co-stimulatory signals to activate T cells [73]. Under malignant conditions, the BTLA/CD160/HVEM axis is associated with an impaired anti-tumor immune response in various tumor entities, as reviewed in detail by Ning et al. [74].

V-domain immunoglobulin suppressor of T cell activation (VISTA) is a novel, rather uninvestigated B7 family checkpoint protein. Recently, it has been shown that V-Set and immunoglobulin domain-containing protein 3 (VSIG-3) binds to VISTA, resulting in T cell inhibition and suppression through a novel VSIG-3/VISTA pathway [75]. Furthermore, VISTA is implicated in a broad spectrum of immune responses and suppresses T cell activation [76]. As a paradigm, the inhibitory immune checkpoint VISTA can act as a ligand and as a receptor on T cells and APCs to attenuate the immune response by suppressing the T cell-mediated immune response [77,78]. The specific physiological role of VISTA remains unclear. Observations in malignant settings suggest that VISTA expression suppresses the T cell-mediated response and promotes immune evasion [79,80,81,82]. Additionally, it was reported that VISTA is mainly expressed on immune cells, especially CD8+ T cells, and is associated with immunosuppressive conditions in various tumor types, including melanoma, acute myeloid leukemia (AML), hepatocellular carcinoma (HCC), small-lung-cell carcinoma, gastric cancer, and colorectal cancer, as reviewed in detail by ElTanbouly et al. [83] and Tagliamento et al. [84].

B7-H3 and B7-H4 are immunosuppressive B7-CD28 family members, which dampen T cell effector function. For both, no corresponding receptor on T cells has been identified so far. Elevated levels of both molecules were observed and correlated with poor clinical outcomes in various tumor entities [85,86].

##### Co-Inhibitory TIM Family Members

The cell-surface molecule T cell immunoglobulin and mucin domain-containing protein-3 (TIM-3) is a key member of effector T cell function and belongs to the TIM family. The TIM family contains two additional members (TIM-1 and TIM-4) in humans. TIM-1 and TIM-4 function as co-stimulatory molecules for T cell activation, whereas TIM-3 possesses an immune inhibitory function. The major function of TIM-3 is to inhibit the Th1 and cytotoxic T cell responses and the expression of TNF and IFN-γ. Hence, TIM-3 is an important negative regulator of adaptive and innate immunity [87,88]. So far, four distinct ligands have been identified to bind to TIM-3: galectin-9, phosphatidylserine (PtdSer), high mobility group box 1 (HMGB1), and carcinoembryonic AG-related cell adhesion molecule 1 (CEACAM1) [89,90,91,92]. Galectin-9 expression is upregulated by IFN-γ contributing as part of a negative feedback loop such as PD-L1 and is expressed by immune cells such as T cells, B cells, macrophages, and mast cells, and by non-immune cells such as endothelial cells and fibroblasts [26,93]. PtdSer is a phospholipid that is exposed on the surfaces of apoptotic cells and interacts as a ligand with all of the TIM family members, but with a remarkably lower affinity [90,94,95]. The interaction of PtdSer and TIM-3 is crucial for the clearance of apoptotic cells, but its role in cancer remains unclear [95]. HMGB1, a DNA-binding protein, is secreted by cancer cells and by other cell types and can interact with other receptors alone or in complex with DNA or LPS [91,96,97]. HMGB1, but not galectin-9 or PtdSer, binding to TIM-3 is responsible for regulating the nucleic acid-mediated innate immune responses [91]. CEACAM1 acts as a self-ligand on T cells. It is expressed at high levels on activated T cells, but not on naïve T cells, and functions as a negative regulator of T cell responses [92,98,99]. CEACAM1 expression has also been identified on macrophages, DCs, monocytes, and tumor cells [100,101,102]. The anti-tumor immunity is inhibited by TIM-3 by mediating T cell exhaustion [103,104,105]. In cancer, TIM3+ PD-1+ CD8+ TILs are associated with the most dysfunctional or terminal stage of CD8+-T cell exhaustion, whereas single positive TIM3- PD-1+ CD8+ TILs exhibit a weaker exhaustion/dysfunctional stage and double negative TIM3- PD-1- CD8+ TILs possess good effector function [25,52,106].

##### Co-Inhibitory Molecules of the Immunoglobulin (Ig) Superfamily

The co-inhibitory receptor lymphocyte activation gene 3 (LAG-3) is a transmembrane protein that is found on the surface of cells and exerts a high structural homology to CD4 [107]. Continuous antigen exposure in tumors or during chronic viral infections leads to high and persistent co-expression of LAG-3 with other co-inhibitory molecules such as PD-1 and TIM3 on T cells, resulting in T cell dysfunction [108]. Tumor-infiltrating T cells are also typically exposed to tumor-associated antigens and consequently express high levels of inhibitory co-receptors such as LAG-3, leading to a state of exhaustion [109,110,111]. To date, several ligands are known to interact with LAG-3, e.g., galactin-3, LSECtin, and MHC class II complex [112,113,114,115,116]. A recent work identified fibrinogen-like protein 1 (FGL1), which is a liver-secreted protein, as an additional immune inhibitory ligand that is highly expressed in human cancer cells [117]. LAG-3 facilitates inhibitory effects on effector T cells and Tregs. In CD8+ T cells, LAG-3 crosslinks with the CD3/TCR complex and inhibits TCR-induced T cell proliferation, the production of cytokines, and calcium influx [118].

The signaling lymphocyte activation molecule F4 (SLAMF4, CD244, 2B4) is a member of the SLAM family of proteins and is also part of the immunoglobulin (Ig) family [99,119,120,121]. SLAMF4 binds to the high-affinity ligand SLAMF2 (CD48), as well as to CD2 with low affinity, causing either co-stimulatory or inhibitory signals depending on the presence of intracellular proteins within the immune cells. When SLAM-associated protein (SAP) is present, co-stimulatory signals are induced. In contrast, when Ewing sarcoma-activated transcript 2 (EAT2) is present, immune inhibitory signals are triggered [122,123,124,125]. In human cancers, SLAMF4 is expressed on exhausted CD8+ T cells and it is co-expressed with other inhibitory receptors [126,127]. SLAMF4 seems to be implicated in immune evasion, but its interplay with other co-stimulatory and/or co-inhibitory signals has not been fully deciphered so far.

##### Co-Inhibitory Nectin and Nectin-Like Binding Receptors

T cell immunoglobulin and ITIM domains (TIGIT, or WUCAM, Vstm3) is a receptor belonging to the nectin and nectin-like molecules (Necls) [128]. TIGIT binds to three ligands, namely CD155 (PVR), CD112 (Nectin-2), and CD113 (Nectin-3), that are expressed on APCs and cancer cells [129,130,131]. The main ligand of TIGIT is CD155 with the highest affinity, and it binds with lower affinity to CD112 and CD113 [129,130,132]. CD155 expression is mainly found on DCs, T cells, B cells, and macrophages, but also in non-hematopoietic tissues [120]. CD112 is expressed in both hematopoietic and non-hematopoietic tissues [133,134]. Analogous to the CTLA-4/CD28 pathway, CD226 (DNAM-1), which is a co-stimulatory counterpart of TIGIT, and CD96 (Tactile), which is a novel inhibitory receptor, compete with TIGIT to bind to CD112 and CD155 in order to fine-tune the immune responses. However, TIGIT binds to CD155 with much higher affinity, followed by CD96 and then CD226 [128,135,136,137]. The TIGIT/CD96/CD226 pathway is more complex than the CTLA-4/CD28 pathway. CD226 can bind CD112 as a ligand and the CD112R (PVRIG), which is another novel immune checkpoint, competes with CD226 and TIGIT to bind to CD112 [128,134,138,139]. Within the TME, the expression of TIGIT is increased in human and murine TILs and is observed in a broad range of malignancies [140,141,142,143,144]. In tumor tissue, TIGIT expression on CD8+ T cells and Tregs plays a major role in driving suppression within the TME [133,140,142]. It was observed that a TIGIT blockade might restore T cell responses in various malignancies [142].

##### Butyrophilins Function as Co-Inhibitory Surface Molecules

Butyrophilin (BTN), butyrophilin-like (BTNL) and the selection and upkeep of intraepithelial T cell-like factors (SKINTL) are a more recently identified class of T cell co-inhibitory/co-stimulatory molecules that share strong structural similarities with the B7 family. They belong to the Ig superfamily that consists of extracellular IgV and/or IgC domains followed by a transmembrane domain and a cytoplasmic B30.2 domain, which was found in most of the members [145,146,147]. In humans, the BTN family consists of seven proteins (BTN1A1, BTN2A1, BTN2A2, BTN2A3, BTN3A1, BTN3A2, and BTN3A3), BTNLs of five proteins (BTNL2, BTNL3, BTNL8, BTNL9, and BTNL10), and SKINTL [148,149,150].

Most of the BTN/BTNLs, which have been characterized, inhibit T cell proliferation and cytokine production through currently unidentified receptors. However, for BTN3A1, BTN3A2, and BTNL8, the co-stimulatory function resulting in T cell activation has been described [148,151,152,153,154].

The number of studies investigating BTN/BTNLs in malignancies is rather limited. It was observed that BTN3A2 expression was associated with the infiltration of CD4+ and CD8+ T cells and a better prognosis in ovarian cancer, indicating its immune co-stimulatory properties [155]. Furthermore, it was shown that BTN3A1 is highly expressed in human high-grade serous ovarian carcinomas, where it inhibits tumor-reactive αβ TCR activation, demonstrating its co-inhibitory effects [156].

##### Intracellular Factors Acting as Negative Regulators of Anti-Tumor Immunity

In addition to surface molecules and soluble factors, intracellular immune checkpoints that modulate T cell function have been identified, namely nuclear receptor subfamily 2 group F member 6 (NR2F6) and poly(rC)-binding protein 1 (PCPB1) [157,158].

NR2F6 possesses a feedback function upon TCR ligation, by repressing the transactivation/DNA binding of NFAT and activator protein 1 (AP-1), which are key transcription factors of TCR stimulation. The DNA-binding affinity of NR2F6 is regulated at the post-translational level by Ser86 phosphorylation mediated by PKC, resulting in diminished DNA binding. It has been demonstrated that NR2F6 mechanistically controls the magnitude and duration of the IL-2, IFN-γ, and TNF-α expression of effector CD4+ and CD8+ T cells, thereby affecting T cell activation and effector outcomes. Data generated from tumor-bearing NR2F-deficient mice that showed reduced tumor growth caused by their hyperactive tumor immunity indicate the key function of NR2F6 as a negative intracellular regulator of the T cell immune response and as a potential target to restore anti-tumor immunity [157].

PCPB1 is an RNA-binding protein, which is upregulated in activated CD4+ and CD8+ T cells upon TCR stimulation. It prevents the conversion of the effector T cells into Tregs, and thereby stabilizes effector function and subverts immunosuppressive conditions. T cell-specific PCPB1 deletion in mouse tumor models caused Treg differentiation and the induction of multiple checkpoint molecules including PD-1, TIGIT, and VISTA on tumor-infiltrating lymphocytes and reduced anti-tumor immunity [158].

#### 3.1.3. Pathways Regulating the Immune Checkpoints and Inhibitory Ligands

The transcriptional control of immune checkpoints in immune cells, as well as of inhibitory ligands in cancer cells, have been understudied so far.

It was reported that the regulators of cell survival and proliferation control the expression of inhibitory ligands, namely members of AP-1, MYC, which is a signal transducer and activator of transcription 3 (STAT3), hypoxia-inducible factor 1-alpha (HIF-1a), and nuclear-estrogen receptor α (ER-α) [159,160,161,162,163]. AP-1 is a dimeric transcription factor that comprises four DNA-binding-protein family members (Jun, Fos, musculoaponeurotic fibrosarcoma (Maf), and activating transcription factor (ATF)) [159,164]. It regulates the inhibitory ligands upon immune cell activation via direct binding of the promotor region, e.g., as observed in Hodgkin’s lymphoma by the binding of Jun to the enhancer region of the PD-L1 promoter, resulting in PD-L1 expression [165,166]. MYC, STAT3, and HIF-1a also possess a binding site in the PD-L1 promoter and seem to induce its expression [160,161,167,168]. ER-α also binds the PD-L1 promoter but represses PD-L1 expression, indicating its function as a negative regulator of inhibitor ligands [162]. These transcription factors are often stimulated when oncogenic-signaling cascades, including mitogen-activated protein kinases (MAPK), JAK/STAT, Wnt, and PI3K/AKT, are activated [164,169,170]. Moreover, a high expression of PD-L1 and PD-L2 is frequently observed because of genetic alterations directly affecting the gene locus of PD-L1 and PD-L2 and/or the JAK/STAT activation that is caused by its amplifications [166,171,172,173].

For immune checkpoints, it was reported that their promoter is bound by transcription factors of the NFAT family (mainly NFAT1 and NFAT2), the AP-1 family, and members of the NR4A nuclear-orphan-receptor family [174,175,176,177,178,179,180,181]. For NFAT1 and NFAT2, it is known that both are responsible for the expression of immune checkpoints such as PD-1, TIM-3, LAG-3, TIGIT, and CTLA-4, as well as exhaustion-associated genes in the absence of AP-1 transcription factors [176,182,183]. For the nuclear receptor subfamily 4A (NR4A), it was observed that two members of this family, namely NR4A2 and NR4A3, are highly expressed in exhausted T cells [182] and those exhaustion-specific accessible regions are enriched for the consensus of binding sites for NR4As in exhausted T cells [184,185]. Furthermore, it was reported that NR4A1, which is the third and last member of the NR4A family, binds the PD-1 promoter [180,181] and that this receptor is preferentially recruited to the AP-1 transcription factor, where it represses T cell effector gene expression by inhibiting AP-1 function [180]. Furthermore, CAR T cells that were lacking all three NR4A members displayed the phenotypes and gene expression profiles of CD8+ effector T cells and chromatin accessibility, which was enriched for the binding of motifs for transcription factors involved in the activation of T cells [181]. Recently, another study supported the central role of TOX proteins in CD8+ T cell exhaustion. TOX is a transcription factor family consisting of TOX1, TOX2, TOX3, and TOX4 [186]. Models causing T cell exhaustion demonstrated that TOX protein was upregulated and remained high in exhausted CD8+ T cells. Additionally, the deletion of TOX promotes the production of IFN-γ and TNF in these conditions [187,188,189,190]. Like NR4As, in a CAR T cell model, TOX1 and TOX2 were highly upregulated in exhausted T cells. The deficiency of TOX1 and TOX2 in CAR T cells decreased the expression of inhibitory receptors and increased effector T cell function. Furthermore, TOX proteins cooperate with NR4As causing T cell exhaustion-mediated calcineurin signaling via NFAT in the absence of AP-1. NFAT, NR4As, and TOXs promote the expression of immune checkpoints [191].

Furthermore, several microRNAs (miRNAs) have been identified to directly bind and downregulate immune checkpoints and their inhibitory ligands, whereas miR-34, miR-200, and miR-570 bind and suppress PD-L1 [192], miR-138 binds and suppresses PD-1 and PD-L1 [192], miR-28 binds and suppresses PD-1 as well as BTLA [193], and miR-155 binds and suppresses the immune checkpoints [194,195]. Additionally, one long non-coding RNA (lncRNA), namely NKX2-1-AS1, has been identified to negatively regulate the PD-1/PD-L1 axis [196].

#### 3.1.4. Soluble Factors Causing Immunosuppressive Condition in the TME

##### 3.1.4.1. The Role of Cytokines in Immune Evasion

Cytokines are small, soluble molecules with pleiotropic effects in normal cells, in host defenses against pathogens, and in the eradication of tumor cells [197] and can be either pro- or anti-inflammatory by nature. They act as major regulators of the innate and adaptive immune systems that allow immune cells to communicate over short distances in a paracrine and autocrine fashion by controlling their proliferation, differentiation, effector functions, and survival [198]. The action of cytokines is usually short-lived and limited; however, dysregulated and chronic cytokine release is a hallmark of auto-immune disease, cancer, and immunosuppression [199]. Cytokines play a key role in tumorigenesis by stimulating intracellular signaling. On the one hand, they cause tumor cell proliferation and/or invasion [200], and the regulation of immune-activation and/or immunosuppressive processes, on the other hand. The immune regulatory functions of cytokines are listed in Table 4. Interestingly, the cytokine signaling is mainly mediated by the JAK/STAT-pathway that is known to control immunosurveillance as well as evasion [201].

##### 3.1.4.2. Pattern Recognition Receptors Signaling

The first line of defense is mediated by the innate immune system. To establish an effective immune response, these innate immune cells use intracellular or membrane-bound pattern recognition receptors (PRRs) to recognize the pathogen-associated molecular pattern (PAMP) or damage-associated molecular pattern (DAMP) molecules that are released from dying cells [223]. PRRs can be classified based on their protein domain homology as toll-like receptors (TLRs), nucleotide-binding oligomerization domain-like receptors (NLRs), retinoic-acid-inducible gene-I-like receptors (RLRs), C-type lectin receptors (CLRs), or cytosolic DNA sensors (CDSs). They are not only expressed on the cell surface but also widely distributed in the intracellular membrane compartment and the cytoplasm. PRRs are mainly expressed on innate immune cells (macrophages and DCs) and B cells as well as, to a lesser extent, on other immune cells such as mast cells, Tregs, monocytes, and basophils. Additionally, they can also be expressed on tumor cells. Under malignant conditions, ligand binding causes the activation of downstream pathways that directly affect tumor cells and/or promote the production and secretion of cytokines, e.g., IFNs, pro-inflammatory factors, and chemokines, in order to activate the adaptive immune system. Thus, PRRs possess regulatory functions of immunosurveillance and tolerance in tumor immunity [224,225,226,227].

TLRs, which are the most studied PRRs in tumors, consist of 13 subtypes, of which TLR11, 12, and 13 are not expressed in humans. Some TLRs (TLR1, 2, 4, 5, 6, and 10) are presented on the cell surface, whereas others (TLR3, 7, 8, and 9) are located in the intracellular compartment [224,225]. Ligand binding (membrane components to surface receptors or nucleic acids to intracellular members) causes NF-κB- and MAPK-pathway activation in either a MyD88-dependent or MyD88-independent manner, which subsequently causes the secretion of pro-inflammatory cytokines and IFNs. TLR signaling is crucial in the activation of adaptive immunity [228,229,230,231,232]. Numerous negative regulators, which suppress TLR signaling and thereby modulate immune responses, can also be found [233]. In the TME, almost all of the immune cell types express TLRs. Thus, TLR stimulation can modulate immune cell function and activation. It can suppress Tregs and/or MSDCs, increase cytotoxic T cell activity, cause switches from Th1 to Th17 responses, lead to M1 macrophage differentiation and increase antigen presentation, suggesting the usage of TLR agonists to promote anti-tumor immune responses [234].

NLRs are intracellular PRRs that comprise five family members (NLRA, NRLB, NLRC, NLRP, and NLRX) and are derived from 22 genes in humans [224,225,226]. Upon activation, NLRs form protein complexes called inflammasomes, whereas others orchestrate NF-κB and MAPK signaling [235]. It has been demonstrated by a few studies that NLR signaling can prime anti-tumor immune responses that are mediated by DCs, macrophages, and T cell activation [236].

RLRs are also intracellular PPRs that mainly include three members: RIG-I, MDA5, and LGP2. They can be activated by double-stranded RNA, resulting in activation of the IRF (IFN-regulatory factors, mainly IRF-1, IRF-3, and IRF-7) transcription factors and the NF-κB pathway [224,225,226]. In the TME, RLR stimulation enhances anti-tumor immunity through the activation of cytotoxic T cells and NK cells, and the inhibition of Tregs [224].

CLRs are expressed on the plasma membrane and can recognize polysaccharides that are present on pathogens and self antigens [225,226]. CLRs are widely expressed on myeloid cells such as macrophages, neutrophils, and DCs. Many CLRs, such as dectin-1, 2, 3, Mincle, and DEC-203, trigger cellular immune responses, which are mainly caused by NF-κB- and MAPK signaling-induced cytokine secretion. In contrast, some CLRs, such as MICL and DCIR, have immune inhibitory effects by controlling the maturation, activation, and proliferation of DCs caused by the inhibition of TLR signaling [225,237].

CDSs are expressed in the cytoplasmic compartment and recognize abnormal DNA. CDSs include IF116, cGAS, AIM2, and DAI. Among them, cytosolic double-stranded DNA promotes the AIM2-mediated inflammasome and enhances the production of pro-inflammatory factors such as IL-1β and IL-18. In contrast, the recognition of double-stranded DNA by cGAS or IF116 causes the activation of STING, leading to the expression of IFN-β and other cytokines such as TNF, IL-1β, and IL-6. Furthermore, it has been demonstrated that cGAS-STING activation in the TAMs and DCs of the TME can enhance anti-tumor immunity [225,226].

##### 3.1.4.3. The Role of Prostaglandin E2 (PGE-2) in Immune Evasion

PGE-2 is one of the most widely produced prostaglandins in the human body. It carries out paracrine and autocrine signaling functions and is involved in multiple physiological processes. However, there is also a large body of evidence implicating PGE-2 and its metabolic enzymes in cancer progression. In tumors, PGE-2 can be produced by the stroma, by TILs, or by cancer cells themselves. High levels of PGE-2 play a role in promoting tumor formation, growth, and metastasis in different types of cancer, including colon cancer, breast cancer, lung cancer, and melanoma [238,239,240]. The reported effects of PGE-2 in tumors are very diverse. They range from apoptosis resistance [241], angiogenesis [242,243], and enhanced migration of tumor cells [244,245] to the suppression of anti-tumor immunity [246]. PGE-2 promotes a TME that contains low numbers of certain infiltrating immune cell types by directly or indirectly affecting different immune cell populations, making it a key disruptor of the anti-tumor immune response [247]. A prominent effect of PGE-2 is the suppression of the conventional type one DC (cDC1s)-mediated immune response. By inhibiting the conventional NK cell (cNK) functions, PGE-2 leads to a lower presence of cDC1 in the TME [248]. cDC1s have been shown to be involved in the recruitment and activation of CD8+ T cells [249,250]. Moreover, they help maintain the cytotoxic responses of CD8+ effector cells in the TME [251]. The sequential failure to recruit cDC1s and CD8+ T cells into the tumor, due to the presence of PGE-2, leads to disruptions at key steps of the cancer immune cycle, such as the antigen presentation, priming, and activation of T cells, and the trafficking and infiltration of T cells into the tumor. PGE-2 in the TME further leads to the differentiation and activation of immunosuppressive myeloid cell species, such as monocyte and neutrophil immunosuppressive phenotypes [252,253,254,255,256], M2 macrophages [257,258], and Tregs [259], thereby further contributing to immune evasion. Zelenay et al. showed in vivo that cyclo-oxygenase (COX) inhibition synergizes with immune checkpoint blockade therapy using anti-PD-1 monoclonal antibodies in a mouse melanoma model [246]. A more recent study could identify a COX-2-associated inflammatory gene signature, which can be used to predict responses to immune checkpoint therapy in multiple cancers in humans [260]. When combined, the effects of PGE-2 result in poorly infiltrated tumors and may therefore lead to immune checkpoint inhibitor therapy resistance.

##### 3.1.4.4. Tumor Metabolism and Its Implication in Immune Evasion

The heterogeneous and dynamic metabolite profile of the TME affects the proliferation and activity of all cellular players involved. Some general features have been ascribed to TME of solid tumors, e.g., hypoxia, scarce nutrient availability, and increased concentrations of waste products, such as lactate, reactive oxygen species, or cysteine [261]. However, these features are spatially unamenable to each part of a single tumor, and to each tumor type, as the TME is highly heterogeneous and some tumors can also rely on oxidative metabolic pathways or can be well vascularized and well supplied with metabolites [262,263,264].

Rapidly proliferating tumors nevertheless possess a highly anabolic appetite and are voracious consumers of glucose [265,266], which enables the ramped-up aerobic glycolysis that is needed for their rapid proliferation [265,266]. Similarly, T cells preferentially use aerobic glycolysis during their clonal expansion into the CD8+ effector phenotype [267,268]. Therefore, the metabolic rewiring occurring in non-transformed, proliferating cells can be attributed to both cancer cells and effector T cells [265,266,269]. Importantly, a specific metabolic profile defines each differentiation state and lineage subtype of T cells, ranging from quiescent, mostly oxidative metabolism in naïve T cells to highly glycolytic metabolism in terminally differentiated effector T cells [270]. Alternatively, as described in the context of acute function and microenvironment, the metabolism of T cells can be graded from hypermetabolic to hypometabolic state [269].

Given their coexistence, cancer cells and T cells are in competition for the nutrients available in the TME. In highly glycolytic tumors, cancer cells tend to outcompete T cells in glucose consumption, thereby inhibiting effector T cell differentiation and functionality [267,271]. Furthermore, other T cell populations, such as CD8+ memory T cells or Treg cells, are less reliant on glucose and are less affected by nutrient competition [267,271]. Therefore, in these conditions, the expansion of immunosuppressive TILs is favored, thus ensuring immune evasion by differentially regulating effector and immunosuppressive TILs. In addition to metabolic disturbance, glucose limitation can also affect effector T cell function through miRNA-regulated epigenetic modifications [272] or by blocking IFN-γ production [267]. Furthermore, the increased aerobic glycolysis of activated T cells allows the maintenance of the Acetyl-CoA pools that are needed for the epigenetic promotion of *IFNG* expression [273], as well as for GAPDH production, which regulates the IFN-γ protein translation [267].

In addition to glucose, the scarcity of other metabolites such as amino acids can also affect T cell activity. For example, the depletion of glutamine, arginine, and tryptophan has immunosuppressive effects that inhibit T cell function and stimulate immune suppressor cell function (e.g., Tregs and/or MDSCs). Arginine is known to induce proliferation, differentiation, and activity of T cells, thus improving their anti-tumor response by promoting oxidative pathways [274]. Tryptophan is needed for protein synthesis in activated T cells, as well as for cell-cycle progression [275], while its catabolites in the kynurenine pathway, such as 3-hydroxyanthranilic and quinolinic acids, can induce selective apoptosis in vitro in Th1 but not Th2 cells [276]. The key players in the depletion of arginine and tryptophan in the TME are arginase and IDO1, respectively, which have both been shown to be upregulated in tumor cells. IDO1 catabolizes tryptophan, thereby depleting tryptophan in the TME, and at the same time produces the oncometabolite kyneurine [277]. In addition to IDO1, IDO2 and tryptophan-2,3-dioxygenase (TDO) regulate the rate-limiting enzymatic conversion of tryptophan in the kyneurine pathway and are therefore being investigated as single or co-targets for the inhibition of the facilitation of the immunosuppressive TME through this metabolic pathway [278,279]. Indeed, a recent publication showed that a novel TDO inhibitor, together with anti-CTLA-4, reduced ectopic colon cancer growth [280].

The waste products of cancer metabolism are also part of the TME and affect immune cell function. The disturbed vascularization that is characteristic of tumors is the major cause of hypoxia and necrosis. Chronic hypoxia [281] and an acidic environment also support immune evasion by inducing T cell differentiation into immunosuppressive Tregs via different mechanisms [282,283]. In highly necrotic areas, dying cancer cells release a high number of cations, specifically potassium, leading to functional starvation of the tumor-infiltrating T cells, which is characterized by autophagy induction and oxidative metabolism, in parallel to downregulation of AKT/mTOR signaling [284,285]. The final outcome is limited T cell effector function and preserved T cell stemness [285].

The interaction of innate immune cells, such as macrophages and NK cells, with the TME is less well examined. However, distinct metabolic features also characterize the two borderline macrophage phenotypes, whereby M1 macrophages have been shown to have enhanced anabolic metabolism, including anaerobic glycolysis, pentose phosphate pathway and fatty acid synthesis, whereas M2 macrophages rely more on catabolic metabolism and oxidative phosphorylation [261].

Results of studies investigating tumor metabolism as a therapeutic target are still diverse and not easily translatable because of the metabolism dynamics and tumor heterogeneity. A balance between immune cell infiltrate, tumor size, and vascularity, as well as the metabolic phenotype, should be taken into consideration before assessing the therapeutic strategy and outcome. For example, even though it seems that nutrient deprivation (applied in the form of fasting, ketogenic, fasting-mimicking diets, etc.) will cause immune suppression, in some cases different findings have been observed. Interestingly, fasting alone reduced acute lymphoblastic leukemia development in mice [286]. Furthermore, fasting-mimicking diets in combination with chemotherapy enhanced the CD8+ T cell-dependent tumor cytotoxicity of chemotherapy in a mouse model of breast cancer [277]. The involvement of the immune system in response to starvation or caloric restriction mimetics has also been demonstrated through the differential regulation of cytotoxic and regulatory T cell activity in the TME [287].

The metabolic profile of T cells is especially important in adoptive immunotherapies. For successful therapy, T cells should display persistence and durability after transfer [270]. There is a possibility to boost these characteristics in ex vivo cell cultures, in order to increase the “quality” of cells for therapy (reviewed in [270]). The metabolic status of the tumor can be predictive of the response to adoptive T cell therapy, as Cascone et al. recently demonstrated that highly glycolytic tumors that were more therapy-resistant [288].

In the TME, a high level of adenosine triphosphate (ATP) is present. ATP is catabolized by CD73, which is an enzyme that is normally expressed by MSDC, TMAs, Tregs, exhausted T cells, and tumor cells in malignant conditions. Thus, CD73 causes high levels of adenosine, which belongs to the group of immunosuppressive metabolites [289]. The high expression of CD73 has been found in many tumor entities and is associated with poor prognosis [290]. Extracellular levels of adenosine are usually low in normal tissue. They increase dramatically upon injury in order to suppress excess inflammation and allow wound healing [289]. These effects are mediated by the adenosine receptors, A2aR and A2bR, which are expressed on multiple immune cells, including T cells, APCs, neutrophils, and NK cells. In these immune cells, both receptors possess inhibitory effects [291].

An elegant experiment using single-cell RNA sequencing provided an overarching picture of the metabolic tumor landscape [292]. Interestingly, the high metabolic heterogeneity of tumors between patients has been observed, while the metabolic profiles of immune cells showed less variability, implying once more the high metabolic plasticity and adaptability of cancer cells compared to immune cells [292]. RNA sequencing could also reveal valuable biomarkers, as exemplified in a recent study showing that the expression of two metabolic enzymes could be differentially correlated to the immune infiltrate in the tumor bed, as well as to patient survival [293].

### 3.2. Altered MHC Class I Expression and Immune Evasion

#### 3.2.1. Antigen Presentation

Antigen presentation via MHC class I and class II molecules is important for activating immune responses [294,295,296]. Nucleated cells synthesize proteins, which are processed and presented as immunogenic peptide-MHC class I complexes on their surfaces. These complexes are recognized by cytotoxic CD8+ T cells, with the purpose of detecting and eliminating altered cells (tumor or virus-infected cells). In contrast, APCs, such as DCs, macrophages and B cells, can ingest antigens, which are processed to be presented in conjunction with the MHC class II molecules to CD4+ T cells.

For the antigen presentation to CD8+ T cells, proteins in the tumor or virus-infected cells need to be proteasomally digested in order to generate short oligopeptides, which are transported from the cytosol to the endoplasmic reticulum (ER) by transporters associated with antigen-processing 1 and 2 (TAP1 and TAP2). In the ER, oligopeptides are loaded onto a nascent MHC class I molecule with the assistance of chaperones. Antigen-loaded MHC class I molecules are then delivered to the cell surface for antigen presentation [297,298].

In humans, classical MHC class I molecules comprise a polymorphic alpha heavy chain that is encoded by the classical HLA genes - *HLA-A*, *HLA-B*, and *HLA-C* - as well as an invariant light chain that is encoded by the *β-2 microglobulin* gene (*B2M*) [295,296]. In addition to the classical MHC class I molecules, there are also non-classical HLA class I molecules. Their heavy chains are encoded by the *HLA-E*, *HLA-F*, and *HLA-G* genes. MHC class II molecules comprise an alpha and a beta chain and are encoded by genes in three different loci (*HLA-DP*, *HLA-DQ*, and *HLA-DR*) [294].

#### 3.2.2. Altered MHC Class I Expression under Malignant Conditions

Antigen presentation on MHC class I molecules is a key component of immunosurveillance, and cytotoxic CD8+ T cells can recognize altered cells through tumor antigens (such as neoantigens, human endogenous retroviral (HERV) antigens, tumor-associated antigens and cancer testis antigens (CTAs)) [299,300,301,302]. Therefore, tumor cells are pressured to develop strategies to evade destruction by the host’s immune system [302]. Altered expression of MHC class I components in order to avoid antigen presentation is common in human malignancies and plays a pivotal role in immune escape [303,304]. The percentage of cases with a low MHC class I phenotype varies greatly depending on the tumor type [305]. Additionally, tumors can exhibit heterogeneous MHC class I expression and the MHC class I expression may change during cancer progression. Moreover, a low MHC class I phenotype is also often associated with a worse prognosis [304].

Altered MHC class I expression can result from hard or soft lesions [306]. Structural genetic alterations, such as mutations affecting the class I heavy chain genes or the *B2M* gene, cause irreversible defects (hard lesions) [295]. In contrast, reversible defects (soft lesions) can derive from alterations in the transcriptional or post-transcriptional regulation of MHC class I antigen presentation pathway components (e.g., genes encoding the class I heavy chain, the invariant light chain and the antigen processing machinery components) [295,305,306]. For example, the repression of gene transcription can be caused by hypermethylated promoter regions of MHC class I antigen presentation pathway genes. MHC class I expression is mediated by NF-κB, IFNs, and NLRC5 [307], and dysregulations in these pathways are also often responsible for MHC class I soft defects [304]. Furthermore, alterations in the regulation of the antigen presentation pathway components may arise from dysregulated miRNAs or lncRNAs [295,305].

Even though tumor cells can escape cytotoxic CD8+ T cells by defective MHC class I expression, NK cells have the capability to destroy these cells via ‘missing-self’ recognition [308,309]. This is due to the inhibitory ligand function of MHC class I complexes, which, when lowered or absent, can result in NK cell activation, leading to cytotoxicity; however, ligands for activating receptors that are upregulated on tumor cells must also contribute to this process [310,311,312]. However, many advanced malignancies exhibit defective MHC class I expression, suggesting that NK-mediated surveillance is circumvented [313]. Hence, tumor cells can employ evading strategies and may also exhibit plasticity to avoid the cytotoxicity of both cytotoxic CD8+ T and NK cells [304,305]. Among others, such strategies include the expression of the non-classical MHC class I HLA-G, which can lead to NK cell inhibition, and the induction of NK cell dysfunction [305,311,312,313,314]. Additionally, NK cell infiltration into solid tumors is often limited and they are mainly confined to the stroma compartments at the tumor invasive margin, also in MHC class I defective tumors [303].

A low MHC class I phenotype represents a major hurdle for effective immunotherapy, as it relies mainly on the cytotoxicity of CD8+ T cells [315]. Indeed, defective MHC class I expression has been associated with a resistance to such regimens [305]. If reversible defects are responsible for the low MHC class I phenotype, then therapeutic drugs could serve to restore MHC class I expression [295]. The clinical benefit of regimens based on NK cells has been shown to be limited so far; however, a number of therapies are currently in development [311,316].

#### 3.2.3. Non-Classical HLA Class I Molecules in Tumors

HLA-E acts as a ligand of the heterodimeric receptor NKG2A/CD94. This receptor is present on circulating NK cells and on T cells with cytolytic function. The interaction of HLA-E and NKG2A/CD94 inhibits this cytolytic function [317,318]. Virally infected cells downregulate HLA-E to favor NK and T cell activation and antiviral responses [319]. In contrast, cancers evade the immune system by the overexpression of HLA-E as well as by recruiting TILs that have strong expression of NKG2A/CD94 [318,320]. High levels of HLA-E were reported in several tumor types, including gynecologic cancers, breast cancer, non-small cell lung cancer (NSCLC), liver, pancreas, kidney, melanoma, prostate, head and neck, stomach, rectal, and colon cancer [321]. The surface expression of HLA-E is associated with functional antigen-processing molecules and cytotoxic T cell infiltration [321]. The blockade of NKG2A enhances the anti-tumor response by NK and T cells. However, the currently available data suggest that monotherapy may be insufficient to achieve anti-tumor effects [318,320].

HLA-F has not been well investigated so far. It was demonstrated that HLA-F can bind to immune inhibitory receptors (ILT2, ILT4, and KIR3DL2), suggesting its potential immunosuppressive function [322]. Under malignant conditions, HLA-F expression seems to be of clinical relevance, as studies demonstrated that it functions as an unfavorable prognostic factor in various cancer types, including NSCLC [323], esophageal squamous cell [324], breast and gastric cancer [325,326]. However, more comprehensive studies are needed in order to fully elucidate its function.

The non-classical MHC class I molecule HLA-G is under non-malignant conditions mainly expressed on extravillious trophoblasts [327,328], which invade maternal tissues during pregnancy, and promotes immune tolerance at the fetal-maternal interface [329,330,331]. Interestingly, many tumors, including colorectal, renal, breast, and lung tumors, have been shown to express HLA-G, which has often been associated with a worse outcome [201,332,333,334,335,336]. In cancer, HLA-G expression can be involved in the evasion from anti-tumor immunity [337]. Indeed, HLA-G has been proposed as a new immune checkpoint in cancer [338]. Similar to other immune checkpoints, a higher expression of pro-inflammatory transcripts appears to be associated with a higher *HLA-G* expression in cancer, suggesting that HLA-G could be upregulated to counteract the host’s immune response [339]. HLA-G exhibits immunomodulatory potential by binding to inhibitory receptors (ILT2 and ILT4) on immune cells, including NK, T, and B cells, as well as macrophages [336]. In addition to direct suppression through binding to inhibitory receptors, further regulatory effects of HLA-G include the induction of immunosuppressive cells such as Tregs as well as the intercellular transfer of HLA-G through mechanisms such as trogocytosis [336,340,341].

#### 3.2.4. The Impact of Epigenetic Mechanisms on Tumor Antigen Presentation

Epigenetic mechanisms, including DNA methylation and histone modifications, which are involved in gene expression, are often dysregulated, as indicated by frequent mutation in DNA-methylating, histone-acetylating and -deacetylating, and methylating genes. These mechanisms are critical for the interactions between tumor and immune cells. It has been demonstrated that tumors commonly hijack these epigenetic mechanisms in order to escape anti-tumor immunity [342,343].

Epigenetic dysregulation contributes to a reduced antigen-presenting function in tumor cells, which enables tumor cells to become invisible to T cells. There are two possibilities to diminish antigen presentation: (1) on the gene expression levels by DNA and/or histone methylation, as well as histone deacetylation, resulting in the low expression of self and tumor antigens (neoantigens and CTAs); (2) on the MHC class I levels (antigen-presenting machinery) by the epigenetic silencing of the genes involved in the antigen-presenting machinery, such as *B2M*, *TAP1*, and *TAP2* [342,343]. Using DNA-hypomethylating agents and inhibitors of histone deacetylase and/or methyltransferase can boost tumor antigen presentation and immunogenicity. In addition, it has been observed that the same epigenetic drugs can induce co-immunostimulatory molecules [344].

It has also been demonstrated that more than 8% of the human genome consists of HERVs arising from retroviruses that infected the human genome millions of years ago [345]. These genomic regions are predominantly silenced by DNA methylation and histone deacetylation and methylations. Treatment with DNA-hypomethylation inhibitors of histone deacetylases and/or methyltransferase restores the expression of HERVs, causing a dsRNA and/or dsDNA sensor, which finally triggers the innate immune response, especially the expression and secretion of cytokines, such as interferons, leading to the enhanced expression of MHC class I molecules on malignant cells [299,342].

#### 3.2.5. Negative Regulation at the Level of APCs in the TME

In the TME, certain subsets of APCs, which hinder immune activation and promote tolerance, are present. These cells are usually immature DCs, expressing low levels of co-stimulatory factors resulting in an insufficient T cell activation [346,347,348]. Additionally, these immature DCs can suppress immune responses by the secretion of anti-inflammatory cytokines such as IL-10 and TGF-β [349]. Furthermore, certain DC subpopulations (CD11c^low^, CD45RB+) and plasmacytoid DCs can induce Tregs [350,351].

Investigations of the DC transcriptional signature revealed that the absence of the transcription factor Rel B, which belongs to the NF-κB family, is associated with increased populations of not only Tregs, but also IL-2-producing memory Th cells [352]. The transcription factor interferon regulatory factor 4 (IRF4) in immunogenic DCs is associated with the IL-10- and IL-33-mediated-differentiation of Th2 (and the priming of Tregs [353]). Furthermore, another transcription factor, namely the dendritic-cell-specific transcript (DC-SCRIPT), promotes IL-10 expression, which is associated with decreased MAPK and enhanced ERK signaling [354], and decreased IL-12 expression [355].

Like tumor cells, DCs can also be involved in the catabolism of the amino acid tryptophan by the expression of IDO (described in detail in Section 3.1.4.4) and thereby promote an immunosuppressive condition in the TME [356,357]. An alternative catabolic pathway causing an immunosuppressive condition in the TME is mediated by TDO (described in detail in Section 3.1.4.4), which can be expressed by specialized myeloid cells [358,359].

### 3.3. Molecules Suppressing the Anti-Tumor Immune Responses beyond Immune Checkpoints

Besides the immune checkpoint components, cytokines, metabolites, and genes that are implicated in antigen presentation, numerous, mostly membrane-bound molecules also exist, that possess immune inhibitory function mainly through their impact on immune cells other than T cells.

CEACAM is a family of proteins including CEACAM1, CEACAM5, and CEACAM6. All three members play a key role in immune modulation: CEACAM1 serves as a ligand of TIM3 (described in detail in Section 3.1). CEACAM5 possesses inhibitory effects on NK-mediated tumor cell lysis, and CEACAM6 dampens myeloid and T cell activation [360,361,362].

Another immunosuppressive gene is the *leukemia inhibitory factor* (*LIF*), which is a crucial factor in embryogenesis. It normally promotes an immunosuppressive microenvironment to protect the embryo from the mother’s immune system. However, it has been shown that it is dysregulated under malignant conditions, favoring immunosuppressive conditions by the inhibition of cytotoxic T cell recruitment to the TME [363].

CD47 serves as a marker of self-recognition. It binds signal regulatory protein α (SIRPα), which is located on macrophages, in order to prompt an antiphagocytic signal [364,365]. In malignant cells it is frequently over-expressed, thereby blocking phagocytosis and favoring metastasis. A high expression of CD47 has been considered as a poor prognostic factor [365].

Semaphorins are a transmembrane protein family that is involved in axonal repair after neuronal injury, cytoskeletal changes, and the migration of endothelial and immune cells [366]. Among this family, SEMA3A, SEMA3B, and SEMA4D (CD100) have all been implicated in the recruitment of TAMs to the TME, and they promote an immunosuppressive microenvironment [367]. SEMA4D binds three types of receptors, Plexin-B1 (PLXNB1), Plexin-B2 (PLXNB2), and CD72, which are all expressed by APCs, endothelial cells, and tumor cells [368]. Upon binding to its receptor, SEMA4D blocks the immune infiltration of active T cells and favors a shift toward Tregs. It also causes macrophage recruitment and differentiation into the M2 phenotype in the TME [368,369].

Common lymphatic endothelial and vascular endothelial receptor-1 (CLEVER-1) is a scavenger receptor that is expressed on endothelial cells and tissues of M2 macrophages [370]. This molecule is implicated in cell trafficking and cellular adhesion and has also been linked with immune modulation, which is mediated by M2 macrophages. Elevated CLEVER-1 levels have been associated with a poor prognosis in certain malignancies. The blockade of the pathway induces an M1 macrophage phenotype in the TME and reactivates and recruits CD8+ T cells [371].

Axl is a tyrosine kinase receptor and a member of the TAM receptor family. It is generally expressed by platelets as well as endothelial, cardiac, hepatic, nervous, and immune cells such as monocytes, macrophages, NKs, and DCs [372]. The binding of Gas6, which is a ligand of Axl, activates phagocytosis and induces an immunosuppressive phenotype in DCs, macrophages, and NK cells [372,373]. Axl also possesses the potential to decrease antigen presentation, increase immunosuppressive cytokines, and indirectly interfere with T cell activation [374].

Phosphatidylserine (PS) is a phospholipid located in the inner layer of the plasma membrane of eukaryotic cells. Once the cell dies, the PS is exposed to the outer layer of the membrane. PS receptors are expressed by endothelial cells, MDSCs, macrophages and DCs, as well as B, T, and NK cells. PS receptors can directly or indirectly bind to PS. The PS/PSR interaction triggers efferocytosis and activates inhibitory pathways that suppress the inflammatory responses to apoptosis [375]. PS is overexpressed by the tumor, endothelial cells, tumor vasculature, and TME. Its levels are further increased by anti-tumor therapies that result in cancer cell death [376]. Targeting this pathway, either by blocking PS or PS receptors, can enhance immune responses against the tumor and potentiate the effects of chemotherapy and radiotherapy [375].

## 4. TME-Targeting Therapeutic Approaches

In past decades, the development of immunotherapies has led to a fundamental improvement in the treatment of solid and hematological malignancies. With the rise of checkpoint inhibitors, hitherto incurable and rapidly evolving NSCLCs or malignant melanoma have changed into chronic diseases with extended prognoses in many cases. Nevertheless, a combination with chemotherapeutic agents is often required to obtain sufficient efficacy. In hematological malignancies, toxic chemotherapies with long-term effects such as the development of secondary malignancies have now been replaced by immunotherapies [377,378,379]. As shown in Table 5, numerous candidates have been chosen to target specific molecules of the TME to restore immunosurveillance.

## 5. Conclusions

The TME is no longer the under-investigated background of malignant cells and has become the main research field in many tumor entities. Based on these findings, novel concepts of cancer development, cancer progression, and points of action for new therapeutic approaches have been developed. Additionally, the role of the immune cells in the TME has been intensively studied. On the one hand, the tumor-promoting function of TILs has been reported, and on the other hand it was demonstrated that TILs are suppressed in their effector function, resulting in the inhibition of anti-tumor immune responses. These immune-evasive processes might be caused by T cell dysfunction induced by tumor and immune cell interaction (e.g., receptor-mediated and/or by secreted molecules) and/or altered antigen presentation. These two affected mechanisms have been investigated in a limited number of tumor entities so far, indicating the need for further comprehensive studies.

The data on the complex interplay of tumor and immune cells are rather limited and are mainly focused on the CTLA-4-CD80-CD86 and PD-1-PD-L1-PD-L2 axes for most of the cancer types. Data on less common co-inhibitory receptors and on secreted molecules are completely missing for most of the malignancies so far. Therefore, detailed investigations aimed at the comprehensive study of the TME and tumor cell interactions are mandatory in order to gain knowledge for the development of novel immune-based therapies to restore an effective anti-tumor immune response.

Collectively, there is an urgent need for genomic studies using a combination of state-of-the-art sequencing techniques (at the single cell level) and imaging-based methods on clinical tumor samples and robust preclinical models to gain knowledge about (1) how these immune-evasive mechanisms are regulated; (2) how these mechanisms can be therapeutically targeted; and (3) which patients can benefit from such therapeutic interventions.

## Figures and Tables

**Figure 1 ijms-22-13311-f001:**
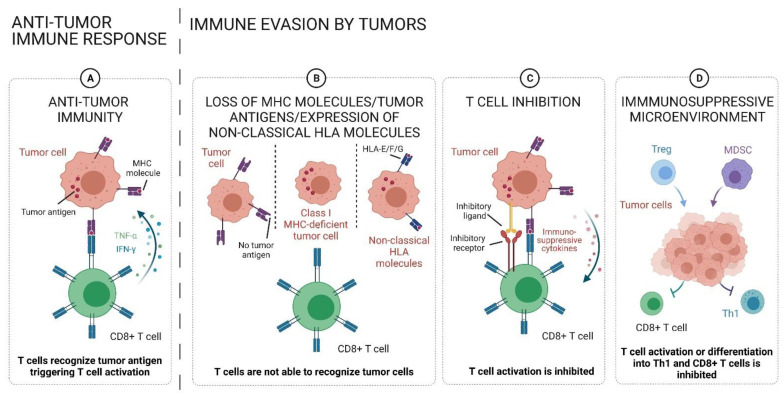
Overview of immune response to tumor vs. immune evasion mechanisms by tumors in the context of CD8+ T cells. (**A**) CD8+ T cells are the preferred immune cells in the role of immunity against targeting cancer through their capacity to kill malignant tumor cells upon the recognition by T cell receptor (TCR) of specific tumor antigens presented on the surface of major histocompatibility complex (MHC) molecules. (**B**) MHC class I molecules can be downregulated on tumor cells and CD8+ T cells are not able to recognize tumor cells. Furthermore, the loss-antigen variant of tumor cells leads to lack of tumor recognition by CD8+ T cells. (**C**) T cell responses are inhibited by the involvement of inhibitory receptors and their corresponding ligands. Immunosuppressive cytokines lead to suppression of the anti-tumor immune response. (**D**) Regulatory T cells (Tregs) suppress the T cell responses to tumors. Myeloid-derived suppressor cells (MDCSs) accumulate and suppress anti-tumor T cell responses. Created with BioRender.com.

**Figure 2 ijms-22-13311-f002:**
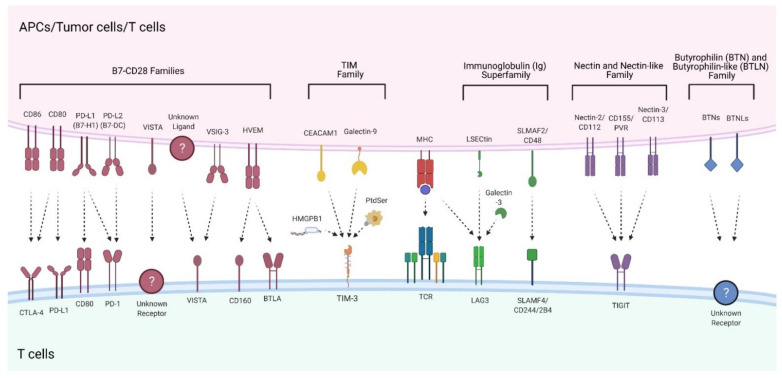
Overview of immune checkpoint molecules/co-inhibitory receptors causing immunosuppressive conditions in the TME. Different immune checkpoint molecules are expressed on T cells and are shown with their ligands expressed on APCs/tumor cells and/or T cells triggering a co-inhibitory signal to suppress effector T cell responses. Immunosuppressive mechanisms are described in detail below. Created with BioRender.com.

**Table 1 ijms-22-13311-t001:** Cells of the innate immune system in TME.

Cell Type	Marker	Production	Function	Reference
Macrophages	CD14+ CD16+ CD64+ CD68+ CD80+ (M1) CD206+ (M2)	M1: IL-12 IL-23 TNF-α M2: IL-10 TGF-β	Macrophages derive from monocytes. They play an important role in host defense against pathogens, stimulation of the adaptive immune system mainly by their function as antigen-presenting cells (APCs), and tissue remodeling. By Th1 cytokines (IFN-γ and/or lipopolysaccharide (LPS)) macrophages differentiate into M1 phenotype and produce pro-inflammatory cytokines. In contrast, macrophages activated by Th2 cytokines (IL-4, IL-10, IL-2) possess the M2 phenotype producing anti-inflammatory factors. Malignant conditions: Tumor-associated macrophages (TAM) can contribute to tumor cell proliferation, invasion, and metastasis, as well as angiogenesis and suppression of T cell-mediated anti-tumor immune responses. TAMs can adopt their differentiation stage in a wide range between M1 and M2 expressing both markers.	[1,6]
Neutrophils	CD11b+ CD15+ CD16+ CD62L+ CD66b+	N1: ICAM1 TNF-α N2: VEGF MMP9	Neutrophils are essential effector cells of the innate immune system. They are the first responders in infection, injury, and defense against pathogens. Malignant conditions: Tumor-associated neutrophils (TANs) can exhibit anti-tumor properties as N1 TANs—mediating cytotoxicity—or pro-tumoral effects as N2 TANs—secreting angiogenesis and invasion promoting factors.	[6]
Eosinophils	Siglec8+ CD193+ CD11b+ CD14- CD62L+	TNF-α granzyme IL-18	Eosinophils are crucial for the control of parasitic infections, bacterial and viral pathogens. Besides, these cells play a central role in inflammation and allergic processes. Malignant conditions: Eosinophils can possess anti-tumorigenic or tumor-promoting functions in different types of tumors. Their different function is mediated by the secretion of anti-tumorigenic or pro-tumorigenic molecules depending on the milieu.	[7]
Mast cells	CD117+ CD203+	VEGF FGF-2	Mast cells represent another important myeloid component of the immune system that contributes to the innate and the acquired immune responses. Malignant conditions: Tumor-associated mast cells (TAMCs) can possess tumor-promoting functions mediated by the secretion of growth- and angiogenesis-promoting factors. In contrast, in some types of cancer TAMCs induce tumor cell apoptosis by IL-4 and TNF-α.	[6]
Myeloid-derived suppressor cells (MDSCs)	CD11b+ CD33+ CD14+ CD15+ CD16+ HLA-DR-	NO ROS iNOS Arginase1 PD-L1 MMP9	MDSCs compromise a heterogeneous immature immune cell population derived from the myeloid compartment. This cell population plays an essential role in the negative regulation of immune responses. Malignant conditions: MDSCs can be induced by GM-CSF, VEGF, and IL-6, which are mainly produced by tumor cells. They can modulate the inflammatory microenvironment via depletion of amino acids and/or via expression of immune inhibitory ligands to inhibit T cell effector function	[6]
DCs	HLA-DR^+^ lineage^−^	IFNs	DCs are the central coordinator of immune response and play a central role in immunity. Their main functions are endocytosis, antigen presentation, and IFN production. Malignant conditions: DCs can play a key role in inducing and maintaining anti-tumor immunity, but in the TME their antigen-presenting function may be inefficient. DCs can differentiate into immunosuppressive regulatory DCs, which limit the T cell activity.	[2]
NK cells	CD3- CD56+	GM-CSF IL-5 IL-8 IL-10 IL-13 CCL2 CCL3 CCL4 CCL5 CXCL10	NK cells belong to the family of innate lymphoid cells with both cytotoxicity and cytokine-producing effector functions. These cells also possess the ability to discriminate target cells, i.e., virus-infected or malignant cells, from healthy cells. This function is based on various cell surfaces consisting of numerous activating and inhibitory receptors. Activating NK cell receptors detect ligands, such as the stress-induced self ligands, infectious non-self ligands, and/or toll-like receptor (TLR), resulting in IFN-γ production and cytotoxicity. Tolerance to self-ligands is mediated by the interaction of the inhibitory receptors and MHC class I molecule. Furthermore, NK cells express the low-affinity Fc receptor CD16, enabling them to exert antibody-dependent cellular cytotoxicity (ADCC). Additionally, NK cells also play a major role in the orchestration of adaptive immune responses by IL secretion. Cytotoxic human NK cells are defined as CD56^dim^CD16^hi^, while immunomodulatory and cytokine-producing NK cells are defined as CD56^bright^CD16^lo^. Malignant conditions: NK cells can directly cause tumor cell lysis; regulate T cell-mediated anti-tumor immune responses by IL secretion and are implicated in the ADCC.	[3]

**Table 2 ijms-22-13311-t002:** Cells of the adaptive immune system in TME: T cell subpopulations.

Cell Type	Marker	Production	Function	Reference
T cells	CD3+	various cytokines	T cells express the TCR complex, which consists of two variable regions—the α- and the β-chains (αβTCR)—in the vast majority of human T cells. The smaller T cell subset—γδ-T cells—just express γ- and δ-chains. Reactive conditions: T cells can recognize foreign or “non-self” material presented as peptides bound to MHC class I or II molecules at the cell surface. Therefore, these cells play an essential role in immune response (bacterial and viral infection via MHC-mediated antigen presentation and tissue/cell graft rejection caused by MHC mismatches). Malignant conditions: T cells can be crucial in anti-tumor immunity initiated by recognition of tumor-specific antigens presented by MHC molecules.	[1,6]
CD8+ T cells	CD3+ CD8+	IL-2 IL-12 Type I IFN granzymes perforin	Reactive conditions: CD8+ T cells mediate immune reactions against pathogens such as viruses and bacteria leading to lysis of infected cells. Malignant conditions: CD8+ T cells can mediate the anti-tumor immune response by recognition of tumor-specific antigens presented by MHC class I molecules leading to tumor cell killing.	[1,6]
CD4+ T cells	CD3+ CD4+	Various cytokines	Reactive conditions: CD4+ T cells are activated by antigens through APCs. These can differentiate into T helper 1 (Th1), T helper 2 (Th2), T helper 9 (Th9), T helper 17 (Th17) cells, follicular helper T cells (Tfh), and regulatory T cells (Tregs). Malignant conditions: CD4+ T cells are less investigated in cancer immunity compared to CD8+ T cells. However, tumor-promoting or anti-tumor immunity was reported for this T cell subtype.	[1,6]
Th1	CD3+ CD4+ STAT4+ T-bet+	IL-2 IL-12 IFN-γ TNF-α	Reactive conditions: Th1 cells secrete IFN-γ to activate macrophages and CD8+ T cells. These cells play an essential role in immunity against intracellular pathogens. Malignant conditions: Th1 cells can promote anti-tumor immunity by activating cytotoxic CD8+ T cells, macrophages, and other APCs.	[1,6]
Th2	CD3+ CD4+ Gata3+	IL-4 IL-5 IL-10IL-13	Reactive conditions: Th2 cells lead to humoral immune responses, typically against extracellular antigens mediated by effector immune cells, including B cells, eosinophils, basophils, and mast cells as well as CD8+ T cells. Malignant conditions: Th2 cells can produce cytokines to downregulate anti-tumor CD8+ T cell-mediated immunity, thereby contributing to tumor growth. This function is particularly mediated by IL-10, which causes inhibition of the DC-mediated antigen presentation and activation of immunosuppressive Tregs. However, Th2-mediated immune responses, such as IL-4 production and activation of eosinophils, can decrease tumor growth.	[1,6]
Th9	CD3+ CD4+ IRF-4+	IL-9 IL-21 IL-10	Reactive conditions: Th9 cells are involved in various pathophysiological conditions of immune response, such as allergic reactions, inflammation and elimination of extracellular pathogens. Malignant conditions: Th9 cells can play a key role in inducing CD8+ T cell-mediated anti-tumor immune responses. Besides, Th9 cells can induce innate cells such as DCs, mast cells, and NK cells to promote a robust anti-tumor immune response.	[1,6]
Th17	CD3+ CD4+ RORγt+	IL-17A IL-17F IL-21 IL-22 CCL20	Reactive conditions: Th17 cells are implicated in immune responses toward bacteria and fungi by recruitment of neutrophils and macrophages. Malignant conditions: Th17 cells and Th17-derived cytokines, such as IL-17, can exhibit anti-tumor and tumor-promoting activity by shaping the TME.	[1,6]
Tregs	CD3+ CD4+ CD25+ FOXP3+	TGF-β IL-2 GITR9 PD-L1 CTLA-4 TIGIT GARP	Reactive conditions: Tregs are specialized to suppress abnormal immune responses to both self and foreign antigens in order to maintain immune homeostasis by inhibiting T cell proliferation and the production of anti-inflammatory cytokines. Malignant conditions: Tregs can suppress anti-tumor immunity, thus promoting tumor development and progression.	[1,6]
γδ- T cells	CD3+ TCRγδ+	IFN-γ	Reactive conditions: γδ-T cells can recognize a broad range of antigens without any presentation via MHC molecules. They can attack target cells directly through their cytotoxic activity or indirectly through the activation of other immune cells. These cells are involved in pathogen clearance, inflammation, and tissue homeostasis. Malignant conditions: γδ-T cells can possess tumor-suppressing function mediated by their own cytolytic properties or activation of other immune cells. However, the tumor-promoting effect, which is mainly mediated by other effector cells, has also been observed in different types of cancers.	[1]
NK-T cells	CD3 CD56 CD4+/− CD8+/−	IFN-γ TNF IL-4 IL-10 IL-13 IL-2	Reactive conditions: NK-T cells rapidly produce large numbers of immunomodulatory cytokines when they are activated and, thus, modulate immune responses against infectious agents, autoantigens, tissue grafts, and allergens. Malignant conditions: NK-T cells can stimulate T and NK cells to eliminate tumor cells during early tumor development. However, in an overstimulated state, these cells can become anergic and differentiate into immunosuppressive NK-T cell subsets, thereby facilitating tumor progression and immune escape.	[1,6]

**Table 3 ijms-22-13311-t003:** Cells of the adaptive immune system in TME: B cell subpopulations.

Cell Type	Marker	Production	Function	Reference
B cells	CD19+ CD20+	various cytokines	B cells are essential players of humoral immunity through antibody (Ab) production. They recognize antigens by the B cell receptor (BCR) composed of membrane-bound antibody. B cells are divided into (1) B1 B cells, mainly found in the peritoneal and pleural cavities; (2) B2 or FO B cells, which are located in lymph nodes; and (3) marginal zone B cells, which are in the marginal sinus of the spleen. The different subsets are activated in a T cell-dependent or -independent way. Reactive conditions: B cell activation begins when the B cell binds to an antigen via its BCR. This activation causes proliferation and differentiation into plasma B cells (PCs), which produce and release antibodies. Malignant conditions: B cells can be implicated in the presentation of tumor-specific antigens to T cells, the production of tumor-specific antibodies and immune regulation as Bregs.	[1,6]
Ab-producing B cells	CD19+ CD20+	tumor-specific IgG and IgA	Reactive conditions: B cells can secrete Abs, and thereby start a specific immune response in viral and/or bacterial infections and autoimmune diseases. Malignant conditions: B cell-mediated Ab production can lead to the killing of tumor cells through the complement cascade activation, phagocytosis by macrophages, and activation of tumor-killing activity of NK cells.	[1,6]
B cells as APCs	CD19+ CD20+ CD21+ CD23+ CD27− IgG1+ CD40+ CD80+ CD86+ MHC class II+	IL-2 IL-6 CCL3 CCL4 ICAM1 GM-CSF	Reactive conditions: B cells can recognize antigen in inflammatory processes in a T cell-independent manner, and they possess the function to present these antigens via their MHC class II surface molecule and thus induce T cell responses. Malignant conditions: B cells can be found nearby T cells in several types of cancers, including when DCs are absent. These cells act as APCs to CD4+ T cells. There exist two forms of this type of cell, causing either anti-tumor immune responses or immunosuppressive intratumoral conditions: (1) the activated B cells (CD69+, HLA-DR+, CD27+, CD21+), possessing Th1 T cell activating function (anti-tumor immune responses); and (3) the exhausted B cells (CD69+, HLA-DR+, CD27−, CD21−), leading to the generation of Tregs (immunosuppressive condition).	[1,6]
Bregs	CD19+ CD21+ CD24+ CD25+ FOXP3+	IL-10 IL-35 TGF-β	Reactive conditions: Bregs are mainly implicated in mediating immune tolerance in autoimmune diseases, including systemic lupus erythematosus (SLE) and multiple sclerosis. Malignant conditions: Bregs possess a tumor-promoting function. They can suppress CD4+ T cell proliferation and induce forkhead box P3 (FOXP3) expression in Tregs mediated by IL-10 and TGF-β. Bregs can also suppress CD8+ T cells in their effector function by IL-10. Furthermore, they cause immune inhibitory receptors PD-L1 on cancer cells through IL-10, IL-35, and TGF-β secretion.	[1,6]

**Table 4 ijms-22-13311-t004:** Cytokines with immune activating or immunosuppressive function.

Cytokine	Function	Reference
*anti-tumorigenic effects—immune activating*		
TNF-α	Enhanced T cell activation Enhanced T cell survival Suppression of Tregs	[202]
IFN-γ	Enhanced antigen presentation by induction of MHC class I Promotion of Th1 together with inhibition of Th2 response, and activation of macrophages	[203]
IL-33	Local increasement of CD8+ T cells and NK cells Increased type I immune response (increased expression of IFN-γ, IL-12 and granzyme B)	[204]
IL-36	Enhanced effector function of CD8+ T cells, NK T cells, and γδ T cells	[204]
IL-12	Enhanced cellular cytotoxicity Enhanced IFN-γ production and differentiation of naïve T cells towards Th1 cells	[205]
IL-2	Expansion of CD8+ T cells by IL-2 Acquisition of effector and memory functions of CD8+ T cells	[206]
IL-18	Activation of CD4+ T cells and/or NK immune responses	[204]
IL-15	Homeostasis and activation of NK cells Expansion and activation of memory T cells	[207,208]
IL-21	Regulation of lymphoid cell, NK cells and myeloid cells	[209]
IL-1	T cell activation Proliferation of B cells	[204]
IL-6	Inhibition of Tregs differentiation Stimulation of Tfh differentiation Production of IL-21 Promotion of differentiation of B cells into IgA secreting plasma cells Involvement in CD28-independent T cell activation	[210]
*pro-tumorigenic effects—immunosuppressive function*		
CSF-1	TAM recruitment and differentiation onto an M2 phenotype	[211]
IFN-γ	Upregulation of indoleamine-pyrrole 2,3-dioxygenase (IDO) and HLA-G, PD-L1 and other immunoregulatory molecules by the JAK/STAT pathway Induction of MDSCs	[203]
IL-18	Regulation of PD-1 by conventional NK cells	[204]
Type I IFN (IFN-α and-β)	Increased expression of TIM-3 and IL-10 Upregulation of CD80 and CD86	[212]
IL-1	Suppression of immune reactions in the TME Recruitment of MDSC and M2 macrophages	[213]
IL-8	Attraction of TAMs, neutrophils, and MSDCs causing suppression of anti-tumor immune responses	[214]
IL-10	Inhibition of cytotoxic effector functions of T cells Interfering with T cell priming Supporting activity of dendritic cells and macrophages	[215,216]
IL-4	Stimulating factors for antigen presenting capacities Promotion of macrophage activation Attraction of M2 macrophages and MDSCs	[217,218]
IL-13	Stimulating factors for antigen presenting capacities Promotion of the macrophage activation Attraction of M2 macrophages and MDSC	[218]
*pro-tumorigenic effects—immunosuppressive function*		
IL-27	Increased expression of immune inhibitory molecules mediated through the transcription factors c-Mac and Prdm1 The key factor for the maximal effector cell expression of PD-L1, LAG-3, CTLA-4, and TIGIT	[219,220]
IL-33	Increased number of immunosuppressive immune cells and innate lymphoid cells	[204]
CCL2	Activation of Tregs and inhibition of T cell effector function	[221]
TGF-β	Inhibition of CD8+, CD4+, NK cells proliferation and cytotoxicity Polarization of macrophages and neutrophils towards a suppressive phenotype Together with FOXP3 driver of Tregs differentiation Together with IL-6 and IL-21 driver of Th17 differentiation	[222]

**Table 5 ijms-22-13311-t005:** Overview of TME targeting therapies.

Target	Drugs	Malignancies	Effect on Immune Response	Phase	Reference *
CTLA-4	Ipilimumab	melanoma NSCLC RCC MSI-H/dMMR CRC HCC MPM MDS AML MPN	anti-CTLA-4 IgG1 mAB	Approved: melanoma NSCLC RCC CRC HCC MPM Phase 1: MDS AML Phase 3: MPN	[380,381,382,383,384,385,386]
	Tremelimumab	melanoma MSTO NSCLC mPDAC PPC mKC MM aHCC	anti-CTLA-4 mAB	Phase 3: melanoma MSTO Phase 2: SCLC HCC mPDAC Phase 1: PPC MM pilot study: mKC	NCT02558894 NCT02485990 NCT02626130 NCT02716805 NCT02519348 Combination with other drugs for many types of cancer (Phase I/II) [387,388]
PD-1	Nivolumab	melanoma SCLC mNSCLC aRCC urMPM, rc/mHNSCC HCC previously treated with sorafenib previously treated a/mTCC, rMSI-H/dMMR mCRC r/r cHL aESCC	anti-PD-1 IgG4 mAB	Approved	[382,383,386,389,390,391,392,393,394,395,396]
PD-1	Pembrolizumab	NCLCS melanoma r/r cHL locally a/m TCC rc/m HNSCC SCLC RCC advanced cervical cancer a/m ESCC ur/ m MSI-H or dMMR CRC	anti-PD-1 mAB	Approved: NCLCS melanoma r/r cHL locally a/m TCC HNSCC Locally a/m ESCC Approved in US:RCC SCLS MSI-H or dMMR CRCadvanced cervial cancer	[397,398,399,400,401,402,403,404,405,406,407,408,409,410,411,412]
	Cemiplimab	aCSCC rc Stage III-IV HNSCC before surgery resNSCLC HCC high risk or locally advanced hormone receptor positive HER2 negative or triple-negative breast cancer mHSPC	anti-PD-1 IgG4 mAB	Appoved: CSCC Phase 2: HNSCC high risk or locally advanced hormone receptor positive HER2 negative or triple-negative breast cancer mHSPC Clinical Trial: resNSCLC HCC	NCT03565783 NCT03916627 NCT04243616 NCT03951831
	Spartalizumab (PDR001)	melanoma metastatic tumors with high PD-1 expression	anti-PD-1 IgG4 mAB	Phase 3: melanoma Phase 2: metastatic tumors with high PD-1 expression	NCT04802876
	Tislelizumab	Rc or a ur/m ESCC rcHCC r/r cHL r/r DLBCL MSI-H/dMMR solid tumors	anti-PD-1 IgG4 mAB	Phase 3: ESCC r/r DLBCLPhase 2: rcHCC r/r cHL MSI-H/dMMR solid tumors	NCT04271956 NCT04663035 NCT04615143 NCT04318080 NCT04799314 NCT04732494 NCT03736889
	Dostarlimab (TSR-042)	aEC a/m CCS LACC melanoma CC NSCLC ovarian neoplasms	anti-PD-1 IgG4 mAB	Approved: aEC: Phase 2: melanoma CCS LACC CC NSCLC ovarian neoplasms	[413,414] NCT04274023 NCT03833479 NCT02715284 NCT04139902 NCT04068753 NCT04655976 NCT03955471 NCT04679064
PD-1	Sym021	advanced solid tumors lymphomas	anti-PD-1 IgG1 mAB	Phase 1	NCT03311412 NCT04641871 [344]
	Camrelizumab	advanced solid tumors NSCLC rcPCNSL r/r cHL	anti-PD-1 IgG4 mAB	Approved in China: r/r cHL Phase 2: resNSCLC PCNSL Phase 1–3: HCC advanced solid tumors	NCT04510610 NCT04342936 [345] NCT04338620 NCT04070040 NCT04564313 NCT04490421
	Toripalimab	melanoma ESCC aNSCLC, r/r DLBCL advanced solid tumors	anti-PD-1 IgG4 mAB	Approved in China: melanoma Phase 3: ESCC Phase 2: r/r DLBCL aNSCLC, ESCC Phase 1/2: DLBCL advanced solid tumors	[415] NCT03829969 NCT04425824 NCT04613804 NCT03811379 NCT04058470 NCT04284488
PD-L1	Atezolizumab	ES-SCLC NSCLC a/m TCC	anti-PD-L1 IgG1 mAB	Approved	[416,417,418]
	Durvalumab	ES-SCLC NSCLC BC TCC PC DLBCL FL CLL	anti-PD-L1 IgG1 mAB	Approved: ES-SCLC Phase 2: BC TCC PC Phase 1/2: DLBCL FL CLL	[409,419,420,421] NCT02401048 NCT02733042
	Avelumab	MCC RCC TCC HL DLBCL	anti-PD-L1 IgG1 AB	Approved: MCC RCC Phase 2: HL Phase 1: DLBCL	[422,423,424,425] NCT03617666 NCT03244176
LAG3	REGN 3767	DLBCL advanced solid tumors	anti-LAG-3 mAB	Phase 1	NCT04566978 NCT03005782 NCT04706715 [426]
	Relatlimab (BMS-986016)	advanced solid tumors melanoma	anti-LAG-3 IgG4 mAB	Phase 1/2: Other Phase 3: melanoma	NCT04080804 NCT03724968 NCT01968109 [427] NCT03470922
LAG3	Sym022	advanced solid tumors lymphomas	anti-LAG-3 IgG1 mAB	Phase 1	NCT03489369 NCT04641871 NCT03311412
TIM-3	MBG453	MDS AML	anti-TIM-3 IgG4 mAB	Phase 2/3	NCT04823624 NCT04150029 NCT04266301
	LY3321367	solid tumors	anti-TIM-3 mAB	Phase 1	NCT03099109 [428]
	BGB-A425	advanced solid tumors	anti-TIM-3 IgG1 mAB	Phase 1	NCT03744468
	TSR-022/ Cobolimab	advanced solid tumors	anti-TIM-3 mAB	Phase 1/2	NCT03680508 NCT04139902
	Sym023	advanced solid tumors lymphomas	anti-TIM-3 IgG1 mAB	Phase 1	NCT03489343 NCT04641871 NCT03311412
	INCAGN02390	advanced solid tumors	anti-TIM-3 IgG1 mAB	Phase 1/2	NCT03652077 NCT04370704
bispecific ab (anti-PD-1/TIM3)	RO7121661	advanced solid tumors	Bispecific antibody against PD-1 and TIM3	Phase 1	NCT03708328
	LY3415244	advanced solid tumors	Bispecific antibody against PD-1 and TIM3	Phase 1	[429] NCT03752177
JAK	Momelotinib	MF NSCLC	JAK1/2 inhibitor	Phase 3: MF Phase 1: NSCLC	[430,431]
	Ruxolitinib	MF PC	JAK2 inhibitor	Approved: MF Phase 3: PC	[432,433] NCT02119663
	Cerdulatinib	PTCL aggressive B-NHL	SYK/JAK inhibitor	Phase 1	[434,435]
	Gandotinib	MPN	JAK2/STAT 3 inhibitor	Phase 2	[436] NCT01594723
	Lestaurtinib	AML MF	JAK2, FLT3 and TrkA Inhibitor	Phase 2: AML Phase 1: MF	NCT00469859 [437,438]
	Pacritinib	MF CRC	JAK2/FLT3 inhibitor	Phase 3: MF Phase 2: CRC	[439] NCT02277093
IL-2	Aldesleukin	mRCC	IL-2 agonist	Approved: mRCC	[440]
	Bempegaldesleukin	advanced solid tumors	IL-2 pathway agonist	Phase 1/3	[441] NCT04410445 NCT04540705
IL-1 and IL-1R3 (IL-1RAP)	Canakinumab (ACZ885)	Early-stage NSCLC	anti-IL-1R3 IgG1 mAB	Phase 2	NCT03968419
IL-1 and IL-1R3 (IL-1RAP)	CAN04	a/m NSCLC a/m CRC a/m BC a/m PDAC	anti-IL-1R3 IgG1 mAB	Phase 1	NCT03267316
IL-8	BMS-986253	advanced solid tumors	anti-IL-8 IgG1 mAB	Phase 1/2	NCT02536469 NCT03400332
Interleukines	TGF-β/Galunisertib	advanced solid tumors	TGF-β Receptor inhibitor	Phase 2	[442,443] NCT02452008
CCL2/CCR2	PF-04136309	mPDAC	CCR2 antagonist	Phase 1	NCT02732938
TLR3	Rintatolimod	advanced solid tumors	Agonist of TLR3	Phase 2	NCT04119830 NCT03899987 NCT03734692 [444]
	Poly-ICLC	PC MM melanomaHCC BC FL B-NHL NSCLC RCC TCC	Agonist of TLR3	Phase 1/2	NCT01079741 NCT02834052 NCT01976585 NCT02643303 NCT02452775 NCT02834052 NCT02061449
TRL4	GLA-SE (GL100)	melanoma FL	Agonist of TLR4	Phase 1/2	NCT02320305 NCT02501473 NCT04364230
TLR5	Entolimod	CRC SCC advanced solid tumors	Agonist of TRL5	Phase 1/2	NCT01527136 NCT01728480 NCT02715882
	Mobilan	PRC	Agonist of TRL5	Phase 1/2	NCT02654938 NCT02844699
TLR7/8	Imiqumod	melanoma various skin cancer CIN PRC BC BCC CL	Agonist of TLR7/8	Approved	[445,446]
	Resiquimod	CL melanoma	Agonist of TLR7/8	Approved	[447,448]
TLR9	CpG7909	CLL B-NHL ESCC	Agonist of TLR9	Phase 1/2	NCT00185965 NCT00669292
TLR2, 4, 9, NLR NOD2	Bacillus Calmette-Guerin (BCG)	BLC	Agonist of TLR2, 4, 9, NLR NOD2	Phase 2/3	NCT03022825
4-1BB (CD137)	Urelumab	advanced solid tumors r/r B-NHL melanoma	anti-4-1BB mAB	Phase 2: melanoma Phase 1: Other	NCT01471210 NCT01775631 NCT02534506 NCT00612664 [449]
CD27	Varlilumab	hematologic malignancies advanced solid tumors	Anti-CD27 IgG1 mAB	Phase 1	[450,451]
CD47/SIRP	Hu5F9-G4 (5F9)	advanced solid tumors r/r B-NHL	anti-CD47 IgG4 mAB	Phase 1: advanced solid tumors Phase 1/2 r/r NHL	NCT02953509 NCT02216409
	ALX148	advanced solid tumors rf B-NHL	blocking SIRPα fusion protein	Phase 1	NCT03013218
	RRx-001	advanced solid tumors lymphomas	molecule that targets CD27/SIRP	Phase 1	NCT02518958
CD73	CPI-006/ Mupadolimab	advanced solid tumors	anti-CD73 mAB	Phase 1	NCT03454451
A2aR	EOS100850	advanced solid tumors	A2AR antagonist	Phase 1	NCT02740985
	AB928/ Etrumadenant	advanced solid tumors	A2AR antagonist	Phase 1	NCT02740985
NKG2A	Monalizumab	rc/m HNSCC	anti-NKG2A mAb	Phase 2	NCT03088059 NCT02643550
LIF	MSC-1	advanced solid tumors	anti-LIF IgG1 mAB	Phase 1	NCT03490669
CSF-1 (M-CSF)/CSF-1R	Lacnotuzumab (MCS110)	advanced malignancies	anti-M-CSF IgG1 mAB	Phase 1/2	NCT02807844
	LY3022855	mBC and mCRPC	anti-M-CSF IgG1 mAB	Phase 1	NCT02265536
	SNDX-6352	advanced solid tumors	anti-M-CSF IgG4 mAB	Phase 1	NCT03238027
	Emactuzumab (RG7155)	advanced solid tumor	anti-CSF1R IgG1 mAB	Phase 1	NCT01494688
	Pexidartinib (PLX3397)	advanced solid tumors a/m PDAC a/m CRC	inhibitor of tyrosine kinase activity of CSF-1R	Phase 1	NCT01525602 NCT02777710 NCT02734433
SEMA4D	Pepinemab (VX15/2503)	aNSCLC	anti-SEMA4D IgG4 mAB	Phase 1/2	NCT03268057
CLEVER-1	FP-1305	advanced solid tumors	anti-CLEVER-1 IgG4 mAB	Phase 1/2	NCT03733990
Axl	Enapotamab vedotin (EnaV)	advanced solid tumors	AXL targeted Antibody-Drug Conjugate (ADC)	Phase 1	NCT02988817
Phosphatidylserine	Bavituximab	a/un HCC	anti-Phosphatidylserine IgG3 mAB	Phase 2	NCT01264705
Imids	Lenalidomide	MM MDS with 5q- indolent lymphoma	inhibitor of ubiquitin E3 ligase cereblon	Approved	[452,453,454,455,456]
Imids	Thalidomide	MM	inhibitor of ubiquitin E3 ligase cereblon	Approved	[456,457]
	Pomalidomide	MM	inhibitor of ubiquitin E3 ligase cereblon	Approved	[456,458]

a: advanced, m: metastatic, r: relapsed, rf: refractory, ur: unresectable, rc: recurrent, res: resectable, r/r: relapsed/refractory, dMMR: deficient mismatch repair, AML: Acute myeloid leukemia, BC: breast cancer, BCC: basal cell carcinoma, BLC: bladder cancer, B-NHL: B cell non-Hodgkin lymphoma, CC: cervical cancer, CCS: clear cell sarcoma, cHL: classical Hodgkin lymphoma, CIN: cervical intraepithelial neoplasia, CL: cutaneous lymphoma, CLL: chronic lymphocytic leukemia, CRC: colorectal cancer, CRPC: castrate-resistant prostate cancer, DLBCL: diffuse large B cell lymphoma, EC: endometrial cancer, ESCC: esophageal squamous cell carcinoma, ES-SCLC: extensive stage small cell lung carcinoma, FL: follicular lymphoma, HCC: hepatocellular carcinoma, HNSCC: head and neck squamous cell carcinoma, HSPC: hormone-sensitive prostate cancer, KC: kidney cancer, LACC: laparoscopic approach to cervical cancer, MCL: mantle cell lymphoma, MCC: merkel cell carcinoma, MDS: myelodysplastic syndromes, MF: myelofibrosis, MM: multiple myeloma, MSI-H/dMMR CRC: unresectable or metastatic microsatellite instability-high (MSI-H) or mismatch repair deficient (dMMR) colorectal cancer (CRC), MPM: malignant pleural mesothelioma, MPN: myeloproliferative neoplasms, MSTO: mesothelioma, NSCLC: non small cell lung carcinoma, PC: pancreatic cancer, PCNSL: primary CNS lymphoma, PDAC: pancreatic ductal adenocarcinoma, PPC: primary peritoneal carcinoma, PRC: prostate cancer, PTCL: peripheral T cell lymphoma, RCC: renal cell carcinoma, SCC: squamous-cell carcinoma, TCC: urothelial carcinoma, WM: Waldenstrom macroglobulinemia * NTC number can be used to identify the clinical study registered at ClinicalTrials.gov, where detailed information on the study can be found.

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
