# Peer review of "Immune Regulatory Processes of the Tumor Microenvironment under Malignant Conditions"

_ijms, 2021, doi:10.3390/ijms222413311_

Round 1

Reviewer 1 Report

This review by Pansy et al., summarizes biology and clinical and advancements in the field of immune regulation of tumor microenvironment.  This review is well written and organized; however I found minor corrections to be made before manuscript is considered for publication.  I have attached a file with correction suggested

Author Response

Thank you for the detailed examination of our work, positive remarks, and for the suggestions for improvement. All suggestions raised by reviewer 1 have been conducted and are highlighted in grey in the manuscript. 

Reviewer 2 Report

The authors provided extensive summary of papers concerning immune regulatory mechanisms in the tumor microenvironment, but most cited papers were published more than 5 years ago. Especially, from lines 1 to 540, less than 20 % cited papers were published for the past five years. Thus, the authors should delete redundant and old literatures, cite the papers which are relevant to the subject, and update the citation. Moreover, in this paper, the authors reviewed papers in a wide variety of fields and as a consequence, the presentation was unfocused and fragmentary. Instead, the authors should give an overview of selected topics, to make the paper more focused. Finally, several sentences are difficult to be understood. For example, lines 97 to 101 and lines 266 to 268. The paper should be extensively edited by a professional editor proficient in scientific English writing, in order to improve the readability substantially.

Author Response

Thank you for carefully examining our work and for the comments. All modifications of the manuscript are highlighted in grey.

According to the reviewer’s concerns, we modified most parts of the review (deleted and re-phrased parts and implemented novel findings) to make it more focused and less fragmented as described below:

  1. As demanded by the reviewer, we updated the literature of the first part and now only cite/report the most relevant findings of more recently published studies.
  2. Furthermore, we have significantly modified section 3, in which T cell function in malignant conditions is described.
    In section 3.1.2 describing the function of immune checkpoints in tumors we removed all redundant parts and focused on the most important finding concerning this topic. Furthermore, we implemented novel findings of recently identified immune checkpoint family, namely butyrophilins, in that section.
    In section 3.2., novel findings on the role of non-classical MHC class-I molecule were added to make this section more focused.
    We also implemented data on molecules possessing an immune-suppressive function in the review because there are already therapies targeting these molecules in clinical trials.
    Finally, novel TME targeting therapies were added to table 5.
  3. To improve the readability and scientific English of the review, several sentences were revised by all authors.

Round 2

Reviewer 2 Report

The authors has modified the manuscript by updating citations and focusing the discussed topics. However, the authors should modify the manuscript to improve the readability.

  1. In the "Toll-like receptor signaling section, the authors should discuss other types of PRRs than TLR.
  2. The authors should number the references sequentially according to their appearance in the main text.

Author Response

We sincerely appreciate your detailed examination of our work and the suggestions for improvement. To improve the readability, we once more checked the spelling and the sentences. . All modifications of the manuscript are highlighted in grey.

As recommended by the reviewer, we implemented novel findings on nucleotide oligomerization domain-like receptors, retinoic acid-inducible gene-I like receptor, C-type lectin receptors, and cytosolic DNA sensors as other types of PRRs and their function in antitumor immunity into the TLR signalling section. To further demonstrate the clinical relevance of PRR, we additionally added their agonists, which are under clinical investigation and/or are clinically approved for antitumor immunotherapy/ies, in table 5.

We would like also to thank the reviewer for pointing out the error in the numbering of references. We carefully checked the whole citation index and corrected it already.